# FinePrompt: Unveiling the Role of Finetuned Inductive Bias on Compositional Reasoning in GPT-4

**Jeonghwan Kim**[1*]   **Giwon Hong**[2*]   **Sung-Hyon Myaeng**[3]   **Joyce Jiyoung Whang**[3†]

[1]University of Illinois Urbana-Champaign    [2]University of Edinburgh    [3]KAIST

jk100@illinois.edu,   g.hong@sms.ed.ac.uk

{myaeng, jjwhang}@kaist.ac.kr

## Abstract

Compositional reasoning across texts has been a long-standing challenge in natural language processing. With large language models like GPT-4 taking over the field, prompting techniques such as chain-of-thought (CoT) were proposed to unlock compositional, multi-step reasoning capabilities of LLMs. Despite their success, the prompts demand significant human effort to discover and validate them. Our work draws attention to the idea of transferring task-specific inductive biases from finetuned models to prompts, as a way of improving GPT-4's compositional reasoning capabilities. To leverage these inductive biases, we formulate prompt templates to ease the transfer of inductive biases. The experimental results on multi-hop question answering and numerical reasoning over text show that our proposed prompt scheme shows competitive zero-shot and few-shot performances compared to existing prompts on complicated reasoning tasks, highlighting the importance of adopting the validated biases of the previous paradigm.[1]

## 1 Introduction

Large language models (LLM) such as GPT-4 (OpenAI, 2023) have demonstrated impressive capability to solve textual understanding problems at a level parallel to or surpassing state-of-the-art task-specific models (Brown et al., 2020; Chowdhery et al., 2022). However, one of the characteristic pitfalls of LLMs is that they exhibit poor zero-shot and few-shot performance in tasks such as multi-hop reasoning (Press et al., 2022) and numerical reasoning over text (Brown et al., 2020; OpenAI, 2023), both of which involve compositional, multi-step reasoning across multiple referents in text.

To overcome such limitation of LLMs, previous works proposed various *elicitive prompting*

---

*Equal contribution

†Corresponding author

[1]Code at: https://github.com/wjdghks950/FinePrompt.git

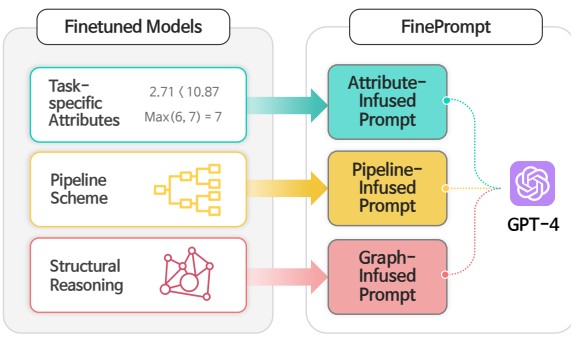

Figure 1: FinePrompt transfers the existing task-specific inductive biases into natural language prompts, guided by the transfer template proposed in this work.

strategies such as Chain-of-Thought (CoT) (Wei et al., 2022), Self-Ask (Press et al., 2022) and Least-to-most Prompting (Zhou et al., 2022). These prompting techniques have effectively unlocked the compositional, multi-step reasoning capabilities of LLMs by generating step-by-step rationales or breaking down an end task into a series of sub-problems. Regardless of their efficacy in improving LLM reasoning, these prompting techniques still entail (i) significant amount of human effort to discover the right prompting strategy, and (ii) lack task specificity that takes into account the characteristic differences between end tasks.

Prior to LLMs and prompt learning, many task-specific finetuned LMs proposed a novel set of inductive biases to improve the compositional reasoning capabilities of finetuned LMs (Min et al., 2019; Groeneveld et al., 2020; Tu et al., 2020; Fang et al., 2020; Ran et al., 2019; Geva et al., 2020; Chen et al., 2020a,b) on tasks like multi-hop question answering (MHQA) (Yang et al., 2018; Ho et al., 2020; Trivedi et al., 2022) and numerical reasoning over text (Dua et al., 2019). For example, NumNet (Ran et al., 2019) injected the strict inequality inductive bias into LMs to significantly improve its performance on DROP (Dua et al., 2019), while DecompRC (Min et al., 2019) divided the multi-hop

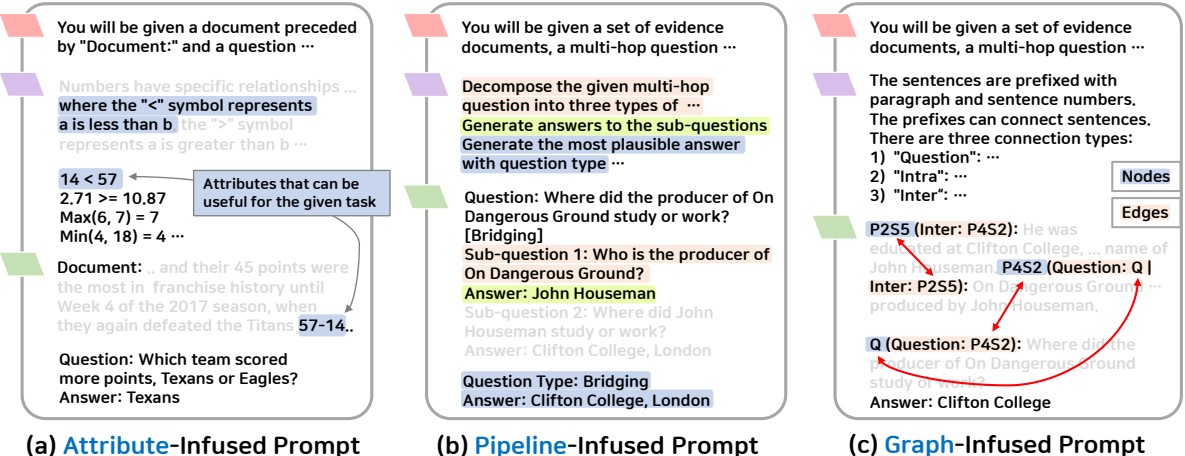

**(a) Attribute-Infused Prompt** **(b) Pipeline-Infused Prompt** **(c) Graph-Infused Prompt**

Figure 2: Illustration of FinePrompt. Each box includes ■ **Task-specific Instruction**, ■ **Finetuned Instruction**, and ■ **In-context Samples & Test Input**. (a) Attribute-Infused Prompt injects a set of end task-specific features. (b) Pipeline-Infused Prompt guides the model to break down a complex end task into a series of subtasks and solve them iteratively. (c) Graph-Infused Prompt infuses the graph's connectivity information within the input text.

questions into a set of decomposed sub-problems to improve the performance on MHQA. However, these finetuning-based methods are difficult to implement in LLMs like GPT-4 because such LLMs are massive in size and their parameters are often inaccessible due to the proprietary nature. In this work, we show that the finetuned models, or more specifically, the set of inductive biases used in such models, can serve as *prompt materials* to improve GPT-4's compositional reasoning as illustrated in Figure 1. Our contributions are threefold; (i) **Reproducibility**[2]: adopting the previously validated finetuned features into prompts to improve the LLM reasoning. (ii) **Systematicity**: providing a template process to turn a finetuned model into a prompt. (iii) **Enhanced performance**: some of the transferred models exhibit strong zero-shot and few-shot capabilities in MuSiQue and DROP.

## 2 FinePrompt

We propose to transfer validated finetuned features into prompts (hence, the name **FinePrompt**) to investigate whether (i) the finetuned features, in forms of prompts, have the same effect of improving the performance of GPT-4 on textual compositional reasoning tasks, and (ii) how the various models/approaches can be effectively transferred to structured prompts. To transfer the features into prompts, we divide models by their properties from

Sections 2.1 to 2.3 as shown in Figure 2.

In each section, we describe which characteristic of a finetuned model aligns with one of the three prompt-infusion strategies; Attribute-Infused (§2.1), Pipeline-Infused (§2.2) and Graph-Infused Prompts (§2.3). Note that inductive biases of finetuned models can be manifested in various forms, because they can be derived from one of the following strategies: (i) integrating additional, task-related features into the models through training (e.g., learning to solve basic arithmetic problems prior to solving complex textual numeric problems (Geva et al., 2020)), (ii) formulating a pipelined process that decomposes a complex reasoning task into a sequential set of sub-problems, and (iii) incorporating external graph structures to leverage the connected, structural inductive bias.

Our work, which aims to transfer the central inductive biases into prompts, directly adopts the previous works (Geva et al., 2020; Ran et al., 2019; Tu et al., 2020; Chen et al., 2020a) to minimize the human effort of extracting the features previously leveraged in these models. For example, as shown in Figure 2, while we have to manually construct the Task-specific Instruction and Finetuned Instruction chunks in the prompts, we can simply adopt the code bases of previous models to extract the necessary features used in In-context Samples.

## 2.1 Attribute-Infused Prompt

Attributes are a set of task-specific features conducive to the end task that provide prerequi-

---

[2]The term *reproducibility* in our work is used to emphasize our contribution that the previously validated fine-tuned ideas can directly be "reproduced" in a form of prompts.

site knowledge. For instance, in order to perform numerical reasoning over text, the model needs to know beforehand how to perform addition/subtraction (Geva et al., 2020) or the definition of strict inequality (Ran et al., 2019) in order to perform a higher-order, compositional reasoning over numbers appearing in text. We define such task-specific features as *attributes* and formulate them as follows. Given a language model $f_\theta(\mathbf{X}; \theta)$, our prompt input $\mathbf{X}$ can be defined as:

$$\mathbf{X} = ([I \parallel P_{attr} \parallel \mathcal{S}_k], x_i) \quad (1)$$

$$\mathcal{S}_k = \begin{cases} \{s_1, s_2, ...s_k\} & \text{if } k > 0 \\ \emptyset & \text{if } k = 0 \end{cases} \quad (2)$$

where $I$ is a **task-specific** and **finetuned instruction** (for full instruction see Appendix D), $P_{attr}$ is the task-specific attribute (e.g., 3 < 11 in NumNet (Ran et al., 2019)), and $\mathcal{S}_k$ is the optional $k$-shot in-context sample from the end tasks' training dataset. Note, $P_{attr}$ can either be in-context sample dependent or not, depending on how the model being transferred used these features. For instance, NumNet leverages the relative magnitude difference between numbers extracted from each sample, whereas GenBERT trains the model on an array of arithmetic tasks (e.g., `19517.4 - 17484 - 10071.75 + 1013.21 = -7025.14`). $x_i$ is the $i^{th}$ end task input. $\parallel$ denotes the concatenation operation. Unlike CoT or Self-Ask, which require manual human annotation of the rationale for the few-shot samples, our prompt simply provides $P_{attr}$ and $s_i$ to the LLM without any manual annotation.

## 2.2 Pipeline-Infused Prompt

Pipelines that break down a complex end task into a series of sub-tasks take the necessary inductive bias (i.e., decomposition) into account. Such biases are especially useful when addressing complicated, multi-hop QA tasks such as MuSiQue (Trivedi et al., 2022). While existing prompting techniques (Press et al., 2022; Zhou et al., 2022) also decompose questions into tractable sub-questions, our pipeline-infused prompts derive directly from existing pipelines implemented by previous works (Min et al., 2019; Groeneveld et al., 2020), reusing the already validated approach as a prompt. The pipeline-infused prompt input $\mathbf{X}$ can be defined as:

$$\mathbf{X} = ([I \parallel \mathcal{S}_k], x_i) \quad (3)$$

where $\mathcal{S}_k = \{c(s_1), c(c_2), ...c(c_k)\}$ and $c$ is the conversion function that converts few-shot samples

into their corresponding pipeline-infused prompt. Note that $c$ includes the decomposition process directly adopted from the existing code base of previous works, providing the decomposed sub-questions, sub-answers and evidences to form $c(s_i)$.

## 2.3 Graph-Infused Prompt

Graphs are often used by finetuned LMs through GNN-based modules (Tu et al., 2020; Chen et al., 2020a) to exploit the connectivity information among textual units (e.g., entities, sentences) that help the LM perform multi-step reasoning. To provide the features conveyed by graphs, we transfer the graph into prompts by identifying nodes within texts and directly inserting an edge preceded by each node as shown in Figure 2(c). Our graph prompt $\mathbf{X}$ is defined as follows:

$$\mathbf{X} = ([I \parallel \mathcal{S}_k], g(x_i)) \quad (4)$$

$$\mathcal{S}_k = \begin{cases} \{g(s_1), g(s_2), ...g(s_k)\} & \text{if } k > 0 \\ \emptyset & \text{if } k = 0 \end{cases} \quad (5)$$

where $g$ is a text conversion function that directly injects node-to-node information into the in-context sample $s_i$ and test input $x_i$. It is worth noting that we do not manually construct the graph or identify nodes present in texts; we directly adopt the graph structures provided by previous finetuned models (Tu et al., 2020; Chen et al., 2020a) and the code bases thereof, an automatic process that does not necessitate manual annotation. The nodes (e.g., sentences) supplied by previous works are directly injected into texts in the form of an indicator token, e.g., P2S5[3], along with the edges which are constructed as proposed in the finetuned models and appended to each node, e.g., connecting sentence nodes based on entity overlap (Tu et al., 2020), or connecting an entity with a number if they appear within a sentence (Chen et al., 2020a).

## 3 Experiments

### 3.1 Settings

**Models** We use GPT-4 as our target LLM in this work, as it is the most powerful LLM on compositional reasoning tasks among other LLMs (OpenAI, 2023). Our experiments are conducted on GPT-4 using OpenAI API, so there is no finetuning of the model involved in this work. We consider the following finetuned models. GenBERT (Geva et al.,

---

[3] P2S5 denotes the fifth sentence in the second paragraph.

| | | Zero-shot | | Few-shot ($k=3$) | |
|---|---|---|---|---|---|
| | | Ans. EM | Ans. F1 | Ans. EM | Ans. F1 |
| Baselines | GPT-4 | 46.41 ±0.29 | 67.90 ±0.32 | 73.20 ±2.27 | 80.50 ±1.45 |
| | Self-Ask | 49.14 ±0.51 | 62.82 ±0.51 | 61.17 ±2.77 | 75.76 ±2.45 |
| | CoT | 69.99 ±0.45 | 81.16 ±0.31 | 65.44 ±4.52 | 77.89 ±2.05 |
| Attribute-Infused Prompt | GenBERT | **77.81** ±0.63 | **84.61** ±0.43 | **77.51** ±2.57 | **83.34** ±1.83 |
| | NumNet | 61.79 ±0.29 | 75.46 ±0.37 | 75.06 ±0.89 | 81.72 ±0.41 |
| Graph-Infused Prompt | QDGAT | 52.73 ±0.66 | 70.36 ±0.42 | 70.86 ±3.17 | 75.58 ±5.58 |

Table 1: Results of FinePrompt on our sampled DROP dev set (256 instances). We provide the averaged score and standard deviation over 5 different iterations, each with a different few-shot sample set.

2020) finetunes on synthetic datasets composed of tasks like addition/subtraction and argmax/argmin. NumNet (Ran et al., 2019) finetunes on number representations to infuse the strict inequality bias into the model. DecompRC (Min et al., 2019) decomposes a multi-hop question into different types of decomposition, generates an answer and evidence per type, and scores them to get the final answer. QUARK (Groeneveld et al., 2020) independently generates a sentence per retrieved paragraph and uses the sentences as context for the final answer. QDGAT (Chen et al., 2020a) is a model with entity-number and number-number graphs to leverage the relationships among numbers and entities. SAE (Tu et al., 2020) is a model with a graph of sentence nodes that uses the sentence-level connections between (and within) paragraphs. Details about the models including CoT and Self-Ask, and hyperparameters are given in Appendix A.

**Datasets** The datasets used in this experiment are a multi-hop QA dataset, MuSiQue (Trivedi et al., 2022), and a numerical reasoning over text dataset, DROP (Dua et al., 2019). Due to the heavy expenses incurring from evaluating on the full evaluation datasets, we sample 256 instances from each dev set as in previous works (Le et al., 2022) and iterate over them for 5 times to address variance. To investigate both the zero-shot and the few-shot performance of the prompt schemes, we test our proposed schemes and baselines along the two axes; the number of few-shot, $k=3$ follows a previous work's setting on DROP (OpenAI, 2023).

**Metrics** The metrics used for DROP (Dua et al., 2019) are answer exact match (**Ans. EM**) and F1 (**Ans. F1**), as the task is essentially a QA that deals with the text match of the generated answer. For

MuSiQue (Trivedi et al., 2022), the task requires the model to perform both the answer generation and the evidence paragraph prediction. To accommodate both tasks and accurately measure whether the generated output sequence contains the answer string, we use answer F1 (**Ans. F1**) and supporting paragraph F1 (**Sup. F1**). The supporting paragraph F1 adopts the implementation by Trivedi et al. (2022).

## 3.2 Results

In Tables 1 and 2, we provide our results on both the zero-shot and few-shot settings over the two compositional reasoning tasks. Our evaluations span two axes: reproducibility of the prompts and their effect on compositional reasoning capability.

**Reproducibility** On both datasets, all our proposed prompts improve markedly over the base GPT-4, demonstrating the same effect the finetuned models exhibit when they incorporate the same inductive biases into their models. Although this work does not exhaustively explore all finetuned models, this result hints at the possibility of incorporating other previously effective inductive biases into prompts to improve the LLM reasoning ability.

**Compositional Reasoning on DROP** As shown in Table 1, attribute-infused prompts, especially GenBERT, excel in both the zero-shot and few-shot settings on DROP. While Self-Ask and CoT improve GPT-4's performance in the zero-shot setting, they show increased variance in the few-shot setting. This provides a stark contrast to the attribute-infused prompts, as they outperform other baselines in the few-shot setting. The graph-infused prompt also improves the numerical reasoning ability, demonstrating that graphs' usefulness, such as

|  |  | Zero-shot | | Few-shot ($k = 3$) | |
| --- | --- | --- | --- | --- | --- |
|  |  | Ans. F1 | Sup. F1 | Ans. F1 | Sup. F1 |
| Baselines | GPT-4 | 62.41 ±0.50 | 82.21 ±0.21 | 65.34 ±0.36 | 75.44 ±0.29 |
|  | Self-Ask | 26.63 ±0.57 | - | 69.50 ±1.20 | - |
|  | CoT | 56.40 ±1.44 | - | 66.33 ±0.73 | - |
| Pipeline-Infused Prompt | DecompRC | **76.67** ±1.04 | **94.18** ±0.62 | **71.74** ±0.71 | **86.70** ±0.57 |
|  | QUARK | 40.17 ±0.74 | 53.73 ±0.31 | 58.53 ±1.64 | 75.60 ±0.62 |
| Graph-Infused Prompt | SAE | 71.90 ±0.64 | 80.00 ±1.36 | 71.11 ±1.54 | 72.75 ±2.76 |

Table 2: Results of FinePrompt on our sampled MuSiQue dev set (256 instances). Self-Ask and CoT do not perform supporting paragraph prediction. We provide the averaged score and standard deviation over 5 different iterations, each with a different few-shot sample set.

connections between different textual units, can effectively be infused to LLMs via prompt.

**Compositional Reasoning on MuSiQue** For multi-hop reasoning on Table 2, both the pipeline and graph prompts outperform other baselines, except for QUARK. The performance drop after applying QUARK's pipeline prompt suggests that, unlike DecompRC which decomposes the question into a series of sub-questions, QUARK independently interprets a paragraph using the multi-hop question, which is not helpful in reducing the complexity of multi-hop reasoning. Moreover, SAE's performance improvement after injecting the graph suggests that even without the lengthy pipeline approach, textual graph prompts better elicit the compositional reasoning ability from the LLM.

## 4 Related Works

**Task-Specific Models** On tasks such as MuSiQue (Trivedi et al., 2022) and DROP (Dua et al., 2019), numerous models have been proposed to enable multi-step, compositional reasoning (Min et al., 2019; Groeneveld et al., 2020; Tu et al., 2020; Fang et al., 2020; Ran et al., 2019; Geva et al., 2020; Chen et al., 2020a,b). These models, prior to the LLM prompt learning paradigm took their place, proposed various novel ideas on this matter.

**Prompt Learning** With prompting techniques like CoT (Wei et al., 2022), Self-Ask (Press et al., 2022) and Least-to-most prompting (Zhou et al., 2022) having shown to improve LLMs' compositional reasoning ability, our work explores how these prompts compare against the FinePrompt scheme. We do not deal with Least-to-most prompt-

ing as it does not deal with compositional reasoning in a textual understanding setting.

## 5 Conclusion

This work studies the transfer of validated inductive biases from finetuned models to prompts and their effectiveness on compositional reasoning of GPT-4. Our empirical results suggest that (i) end task-related attributes and graphs help elicit robust multi-step reasoning capability from LLMs, and (ii) previous finetuned model pipelines, if they involve decomposing a task into a smaller sub-problems, are also effective for prompting. Our work suggests that more can be exploited from the previous pretrain-then-finetuned models and our proposed template can incorporate those features seamlessly into LLM prompting paradigm. We hope future works can explore further in this direction and easily leverage the power of LLMs with FinePrompt.

## Limitations

**Limited Dataset Size** Using GPT-4 for our study incurs substantial cost because of its price ($0.03 per 1,000 tokens), which led us to randomly sample 256 instances from the evaluation sets of both datasets instead of evaluating on the full dataset. Following other previous works (Bai et al., 2023), we deal with reduced dataset size to address the high-cost of using OpenAI GPT model APIs.

**Symbolic Compositional Reasoning Datasets** While our work deals with compositional reasoning datasets within the textual understanding environment, there are other tasks like last-letter concatenation task (Wei et al., 2022) and action sequence prediction such as SCAN (Lake, 2019). However,

they do not deal with textual context understanding. Future works may explore other models on these end tasks as well.

**Extension to other LLMs** While there are other LLMs available such as Alpaca (Peng et al., 2023) and Vicuna (Chiang et al., 2023), which are LLAMA-based (Touvron et al., 2023), instruction-tuned models, they use GPT-generated instruction data to finetune their models. We also note that such LLMs are too compute-heavy to train in our local environment.

**Additional Finetuned Models** We are aware that there are numerous pretrain-then-finetuned LMs for MHQA and DROP. Nevertheless, since we cannot exhaustively consider every single model that has been proposed to date, we select a few models that share commonalities as in Sections 2.1 to 2.3 to investigate their impact on LLMs as prompts.

**Manual Annotation** Manual annotation is unavoidable in FinePrompt since it requires a human annotator to understand the central inductive bias of a model and translate them into textual prompts. Nonetheless, one of the main contributions of this work, which is to reduce the human effort in searching for an effective prompting strategy like CoT (Wei et al., 2022) and Self-Ask (Press et al., 2022) by transferring previously validated inductive biases, holds regardless of the manual effort. Moreover, FinePrompt adopts the code and data bases of previous finetuned models to further mitigate human effort of extracting the inductive bias features.

## Acknowledgements

This research was supported by an NRF grant funded by MSIT 2022R1A2C4001594 (Extendable Graph Representation Learning) and an IITP grant funded by MSIT 2022-0-00369 (Development of AI Technology to support Expert Decision-making that can Explain the Reasons/Grounds for Judgment Results based on Expert Knowledge).

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

## A Additional Settings on Experiments

**Hyperparameters & Datasets** We explain detailed settings such as the hyperparameter setting of GPT-4 and changes to the existing data settings to accommodate the LLM. GPT-4 hyperparameters used in this work is as follows: the temperature of GPT-4 is set to 0.0 to offset randomness during generation. For MuSiQue, there are originally 20 question-related documents for each QA instance. However, due to the context length limit (8K) of GPT-4 and delayed API response, we consider 5 documents; all of which are randomly sampled and consist of 2 gold documents (we adopt the 2-hop question tasks from Press et al. (2022)) and 3 non-gold documents. This setting is unanimously applied to all baselines. For DROP, there is no additional change to the experiment setting as it deals with a single paragraph and a question.

**Models** We adopt the Self-Ask (Press et al., 2022) and CoT (Wei et al., 2022) prompt techniques directly from their papers. While experiments conducted by Press et al. (2022) do not take into account the provided question-related documents (contexts) from MuSiQue (Trivedi et al., 2022), for the fairness of comparing the effectiveness of prompts, we report in Table 2 the results of prompting with the contexts.

For QDGAT, as the official code on the Github repository of the model was not replicable, we implement the QDGAT graph generation with Stanza (Qi et al., 2020); it is used to extract entities and numbers from each document. Note that "DURATION" number type is removed in the process as Stanza does not support the number type. This leaves 7 number types (NUMBER, PERCENT, MONEY, TIME, DATE, ORDINAL, YARD). Moreover, in order to adopt the two essential connections from the QDGAT graph, entity-number and number-number edges, without having to modulate a prolonged text, we denote number-number edges as a group after `NUMBER-NUMBER:` (e.g., `NUMBER-NUMBER: YARD`).

On NumNet, we note that while it uses GNN to finetune number representations, it does not use the connectivity inductive bias as in other graph leveraging models like SAE and QDGAT. Therefore, we add number-specific features of NumNet as Attribute-infused prompt, not Graph-infused prompt.

|  | Zero-shot | Few-shot ($k=3$) |
|---|---|---|
| Self-Ask | 26.63 ±0.57 | 69.50 ±1.20 |
| w/o context | 10.95 ±0.26 | 37.85 ±0.32 |
| CoT | 56.40 ±1.44 | 66.33 ±0.73 |
| w/o context | 27.36 ±0.21 | 36.84 ±1.78 |

Table 3: Additional results on the Self-Ask and CoT performance with and without providing the question-related documents in our sampled MuSiQue dev set.

|  | EM | F1 |
|---|---|---|
| GenBERT finetuned | 59.38 | 71.48 |
| GenBERT 0-shot | 77.81 | 84.61 |
| GenBERT 3-shot | 77.51 | 83.34 |
| SAE finetuned | 55.61 | 54.76 |
| SAE 0-shot | 71.90 | 80.00 |
| SAE 3-shot | 71.11 | 72.75 |

Table 4: Additional results on the original finetuned GenBERT on DROP and SAE on MuSiQue against our FinePrompt counterparts to compare the original finetuned models and FinePrompt. Each FinePrompt variant is denoted by the number of $k$-shot samples used.

**Few-shot In-Context Samples** For our **Few-shot** ($k=3$) setting in Tables 1 and 2, we random sample $k$ instances from the training datasets of DROP and MuSiQue for 5 times. With a total of 15 randomly sampled instances, we manually construct $k$-shot in-context samples for CoT and Self-Ask as both requires humans to provide an intermediate rationale to a question in each sample.

## B Additional Experiments: FinePrompt and finetuned Models

While our work seeks to investigate the effectiveness of the validated, finetuned inductive biases in the form of prompts, we provide additional experiments on how the finetuned models used in this work fare against their FinePrompt counterparts. We have conducted additional experiments with GenBERT on DROP and SAE on MuSiQue to compare the original finetuned models and FinePrompt (shown in Table 4). The results demonstrate that our FinePrompt scheme outperforms its original finetuned counterparts, exhibiting the potential to understudy the finetuned models in a low-resource setting by leveraging LLMs.

## C  CoT and Self-Ask without Contexts

In Press et al. (2022), the base settings of Self-Ask and CoT do not take question-related documents (contexts) into account on the multi-hop question answering task, and do not perform supporting evidence paragraph prediction either; they use their parametric knowledge to decompose the given questions and generate answer rationales. However, as our models applying FinePrompt require contexts provided by documents, we present the performance of CoT and Self-Ask with contexts for a fair comparison in Table 2.

To evaluate how the previous elicitive prompting strategies perform in our experimental setting when contexts not provided, we provide additional experiments in Table 3. Providing the question-related documents shows a substantial increase in both Self-Ask and CoT, notably in the few-shot setting.

## D  Full Prompts

Here we provide the actual prompts used in our work. Each prompt, divided into three distinct groups (Attribute, Pipline, Graph), consists of the following format: (i) Task Instruction, (ii) Fine-tuned Instruction, (iii) In-context Samples & Test Input. For the Attribute-Infused Prompt, we also inject the Input-related Attributes (see Figure 2 for details). At the end of each instruction, the few-shot samples (optional in case of zero-shot) and the end task query (Question:) will be appended for GPT-4 to responsd to.

In the following, the DROP task instructions will be denoted by blue boxes, whereas the MuSiQue task instructions will be denoted by red boxes. The baselines like Self-Ask (Press et al., 2022) and CoT (Wei et al., 2022) are denoted by green as their zero-shot setting shares the same prompt in both datasets.

---

**Base Instruction for DROP (Dua et al., 2019)**

You are a question answering machine that answers a question based on a given document. You will be given a document preceded by "Document:" and a question preceded by "Question:". When you generate the answer, simply generate the answer after "Answer:"

Document: ...
Question: ...
Answer: ...

---

**Instruction for GenBERT(Geva et al., 2020)**

You are a question answering machine that answers a question based on a given document. You will be given a document preceded by "Document:" and a question preceded by "Question:". When you generate the answer, simply generate the answer after "Answer:".

You will also be given a set of related task examples to help you acquire the necessary knowledge to answer a given question based on the document.

Document: ...
Question: ...
Answer: ...

Related Examples:
1) 19517.4 - 17484 - 10071.75 + 1013.21 = -7025.14
2) most(1072.1, 17938, 5708.65, 14739.16) = 17938
3) argmax(toppy 8105.5, cockney 7111.0, nickelic 1463.16, tiredom 6929) = toppy
4) most recent(July 16, 134; June 23, 134; 24 July 134; 28 October 134) = 28 October 134
5) difference in days(April 21, 1381; 13 April 1381) = 7
6) percent not photochemist, floodgate, retiringly :: photochemist 0.82%, morningward 54.4%, floodgate 2.0%, reline 0.78%, retiringly 42% = 55.18

7) Document: "The commander recruited 16426 asian citizens and 15986 asian voters.
The commander borrowed 7 foreign groups from the government. The government passed 3 foreign groups to the commander."
Question: How many foreign groups did the commander recruit?
Answer: 10

## Instruction for NumNet (Ran et al., 2019)

You are a question answering machine that answers a question based on a given document. You will be given a document preceded by "Document:" and a question preceded by "Question:". When you generate the answer, simply generate the answer after "Answer:"

Numbers have specific relationships as shown in the following examples, where the "<" symbol represents "a < b" (a is less than b), the ">" symbol represents "a > b" (a is greater than b), and the "=" symbol represents "a = b" (a is equal to b):

Document: ...
Question: ...
Answer: ...

5 < 6
10 > 6
117 > 25
978 < 979
0 = 0
1.6 < 7.2
9.0 > 8.9
2.6 < 2.9

## Instruction for QDGAT (Chen et al., 2020a)

You are a question answering machine that answers a question based on a given document. You will be given a document preceded by "Document:" and a question preceded by "Question:". When you generate the answer, simply generate the answer after "Answer:"

Some entities and numbers in the provided document can have special connections.
There are a total of two connection types.
1) "ENTITY-NUMBER": Connections between entity and number in the same sentence.
2) "NUMBER-NUMBER": Connections between numbers of the same type. A NUMBER-NUMBER connection is represented by specifying the corresponding number type.

Document: ... ENTITY1 (ENTITY-NUMBER: NUMBER1) ... NUMBER1 (ENTITY-NUMBER: ENTITY1) ... NUMBER2 (ENTITY-NUMBER: ENTITY2, ENTITY3 | NUMBER-NUMBER: YARD)

Question: ...
Answer: ...

**Base Instruction for MuSiQue (Trivedi et al., 2022)**

You are a question answering assistant. You will be given a set of evidence paragraphs, a multi-hop question and you will be asked to do the following:
1) You will read a list of paragraphs (P1, P2, ..., PN) and a multi-hop question ("Question:").
2) You should give the paragraph id you used to derive the answer after "Evidence:".
3) You should provide the answer to the multi-hop question after "Answer:".

Paragraphs: ...
P1: ...
P2: ...
...
PN: ...
Question: ...
Evidence: Pi, Pj, ...
Answer: ...

**Instruction for DecompRC (Min et al., 2019)**

You are a question answering assistant. You will be given a set of evidence paragraphs, a multi-hop question and you will be asked to do the following:

First, decompose the given multi-hop question ("Question:") into all three different versions of single-hop, sub-question sets ("Sub-question 1:", "Sub-question 2:"). The three different question types are as follows:

1) Bridging Type: requires finding the first-hop evidence for Sub-question 1 to find the evidence to answer Sub-question 2.
2) Intersection Type: requires finding an entity that satifies two independent conditions of the two Sub-questions.
3) Comparison Type: requires comparing the property of two different entities in the Sub-questions.

Then, given a question, generate the sub-questions, the corresponding answer and the evidence paragraph ids for each sub-question in the following format:

Paragraphs:
P1: ...
P2: ...
...
PN: ...

Question: ...

[Bridging]
Sub-question 1: ... | Sub-question 1 Answer: ... | Evidence: Pi, Pj, ...
Sub-question 2: ... | Sub-question 2 Answer: ... | Evidence: Pi, Pj, ...

[Intersection]
Sub-question 1: ... | Sub-question 1 Answer: ... | Evidence: Pi, Pj, ...
Sub-question 2: ... | Sub-question 2 Answer: ... | Evidence: Pi, Pj, ...

[Comparison]
Sub-question 1: ... | Sub-question 1 Answer: ... | Evidence: Pi, Pj, ...
Sub-question 2: ... | Sub-question 2 Answer: ... | Evidence: Pi, Pj, ...

Using the previously generated information about the sub-questions, the answers and evidence paragraphs, generate the most plausible answer to the question ("Question:") after "Answer:", and also generate which question type your answer is from as follows:

Question Type: ...
Answer: ...

## Instruction for QUARK (Groeneveld et al., 2020)

You are a question answering assistant. You will be given a set of evidence paragraphs, a multi-hop question and you will be asked to do the following:

1) You will read a list of paragraphs (P1, P2, ..., PN) and a multi-hop question ("Question:").
2) Find one question-related sentence for each paragraph ("Paragraph:") and write that sentence id after "Evidence Sentences:".
3) Read the given set of sentences after "Evidence Sentences for Pi:", where "i" refers to the paragraph id. This set of predicted sentences will serve as your new context to help you answer the question.
4) You should provide the answer to the multi-hop question after "Answer:".

Paragraphs: ...
P1: ...
P2: ...
...
PN: ...

Question: ...

Evidence Sentences for P1: Si
Evidence Sentences for P2: Sj
...
Evidence Sentences for PN: Sk

Answer: ...
Evidence Paragraphs: Pi, Pj, ...

## Instruction for SAE (Tu et al., 2020)

You are a question answering assistant. You will be given a set of evidence paragraphs, a multi-hop question and you will be asked to do the following:

1) You will read a list of paragraphs (P1, P2, ..., PN) and a multi-hop question ("Question:").
2) You should provide the answer to the multi-hop question after "Answer:".
3) You should give the paragraph id you used to derive the answer after "Evidence:".

The provided paragraphs and sentences within are prefixed with paragraph numbers and sentence numbers. For example, the prefix "P2S1" indicates the 1st sentence of the 2nd paragraph.
Also, if sentences are related to other sentences, prefixes can connect them to each other in some form of connection. There are a total of three connection types:
1) "Question": Connections between sentences that are related to the question.
2) "Intra": Connections between sentences within the same paragraph.
3) "Inter": Connections between sentences that are related but belong to different paragraphs.

Paragraphs:
P1S1 (Inter: P2S2 | Intra: P1S2): ...
P1S2 (Intra: P1S2): ...
P2S1 (Intra: P2S2): ... P2S2 (Question: Q, P3S2 | Inter: P1S1 | Intra: P2S1): ...
P3S1 (Intra: P3S2): ... P3S2 (Question: P2S2 | Intra: P3S1): ...
...
PNS1 (Question: Q | Intra: PNS2): ...
PNS2 (Intra: PNS1): ...
Q (Question: P1S1, PNS1): ...
Answer: ...
Evidence: P1, P3, ...

## Instruction for Self-Ask (Press et al., 2022)

You are a question answering assistant.
1) You will be given a question ("Question:").
2) Figure out if any follow up question is needed ("Are follow up questions needed here:") with "Yes" or "No" answer.
3) For each follow up question, give the corresponding "Intermediate answer:".
4) When you generate the final answer, generate the answer after "So the final answer is:".

Question: ...
Are follow up questions needed here: ...
Follow up: ...
Intermediate answer: ...
So the final answer is: ...

## Instruction for CoT (Wei et al., 2022)

You are a question answering assistant.
1) You will be given a question ("Question:").
2) Generate an explanation for your answer after "Answer:".
4) When you generate the final answer, generate the answer after "So the final answer is:".

Question: ...
Answer: ...
So the final answer is: ...