# OpenReview forum: "FinePrompt: Unveiling the Role of Finetuned Inductive Bias on Compositional Reasoning in GPT-4"
_EMNLP/2023/Conference — EMNLP 2023 Findings_

### Official Review · Reviewer_wpwG · 2023-07-31

**Soundness:** 2

**Excitement:**

3: Ambivalent: It has merits (e.g., it reports state-of-the-art results, the idea is nice), but there are key weaknesses (e.g., it describes incremental work), and it can significantly benefit from another round of revision. However, I won't object to accepting it if my co-reviewers champion it.

**Paper Topic And Main Contributions:**

This paper explores the idea of transferring task-specific inductive biases from fine-tuned models to prompts, thereby reducing the human effort required for annotation and infusing task-specificity. Prompt templates are formulated to ease the transfer of inductive biases and the proposed scheme shows competitive zero-shot and few-shot performances compared to existing prompting strategies.

**Questions For The Authors:**

A. How are inductive biases obtained? Is it task-specific or instance-specific? Do inductive biases differ across different fine-tuned models for the same task? If not, this prompting strategy seems to be benefiting from some intermediate annotation of the task itself, rather than having anything to do with fine-tuning.
B. Does the Pattr you obtained for the numerical reasoning task vary depending on which few-shot examples are being presented to the model?
C. In section 2.3 (Graph-Infused prompting) how are the nodes within the text identified? Are they manually identified or is this part of the dataset?
D. How are pipeline-infused prompts different from existing prompting strategies like CoT, self-ask, etc.? It is unclear what you mean by "..derive directly from existing pipelines, reusing the already validated approach as a prompt.."
E. Is fineprompting done on GPT-4 out-of-the-box or one that's fine-tuned for each task?
F. As per your contribution list, can you explain how this method is more reproducible than others?


**Reasons To Accept:**

This work presents some interesting insights into how infusing task-specific inductive biases into prompts can be compelling. Furthermore, it bridges the gap between fine-tuning and prompting, a vital direction given the scale and inaccessibility of present-day LLMs.

**Reasons To Reject:**

The paper lacks clarity in different places. First of all, it is unclear to the reader what prompt-infusion method (Section 2) is to be used for which fine-tuned model properties. Are these properties attributable to the model, training objective, or the task itself? Secondly, it is unclear what is being used to extract the inductive biases that are infused into the prompts. How do the authors identify such information and what makes it immune to the pitfalls of manual annotation?

In summary, I believe this paper could include more details on how fine-tuned models are differentiated and inductive biases are extracted from them. I presume it would be more comprehensible on conversion to a long paper.

**Reproducibility:**

3: Could reproduce the results with some difficulty. The settings of parameters are underspecified or subjectively determined; the training/evaluation data are not widely available.

**Reviewer Confidence:**

3: Pretty sure, but there's a chance I missed something. Although I have a good feel for this area in general, I did not carefully check the paper's details, e.g., the math, experimental design, or novelty.

**Typos Grammar Style And Presentation Improvements:**

typos: 518, 526
It would be useful to include a table that delineates which model/dataset can be attributed to which of the three prompt infusion strategies introduced in this paper.

---

> ### Author Rebuttal · Authors · 2023-08-29
>
> Thank you for your valuable time and feedback for our work. We would like to open ourselves up for discussion about the core contribution of this research.
>
> >**“It is unclear to the reader what prompt-infusion method (Section 2) is to be used for which fine-tuned model properties.”**
>
> Thank you for pointing this out. In the following points we would like to further elaborate on the associations between the prompt-infusion methods and their respective fine-tuned model properties. Although each method and its corresponding model property is elaborated throughout Sections 2.1-2.3, we could not explain in full detail due to the limited space. We will address your concern in the additional space of our final paper by refining our writing to clarify the method-wise associations to fine-tuned model properties.
>
> - As noted in lines 91-98, Attribute-infused Prompts are based on models that exploit task-related attributes that are conducive to improving reasoning on end tasks. For example, knowing the relative magnitude differences such as  3 < 5 is essential to solve arithmetic problems like 5 - 3.
> - For Pipeline-infused Prompts, the associated fine-tuned models should decompose a complex end task (e.g., multi-hop questions) into simpler subtasks (lines 116-123).
> - Graph-infused Prompts are for models that explicitly leverage graph’s structural information (e.g., using GNNs) along with the fine-tuned language models (lines 130-138).
>
> >**“Are these properties attributable to the model, training objective, or the task itself?”**
>
> The fine-tuned properties are attributed to the model since we are seeking to exploit the useful inductive biases derived from the model architecture. For instance, QDGAT (Chen et al., 2020a) uses an external GNN to leverage the connectivity inductive bias, from which we derive our graph-infused prompts. NumNet (Ran et al., 2019), also one of the models studied in this work, incorporates the relative magnitude differences between two numbers (e.g., 3 < 5) as the central inductive bias in order to improve its numerical reasoning performance on DROP.
>
> >**“It is unclear what is being used to extract the inductive biases that are infused into the prompts. How do the authors identify such information and what makes it immune to the pitfalls of manual annotation?”**
>
> The fine-tuned inductive biases, prior to the prompt learning paradigm, were infused into the models through manual, human labor (i.e., code). In accordance with the prompt learning paradigm, our work aims to leverage the same fine-tuned inductive biases in a prompt format while avoiding the time consuming yet useful fine-tuning steps. Furthermore, our work adopts the existing code base of the models dealt in this work, relieving our effort of having to extract the central inductive bias features by hand (e.g., the entities used in SAE are directly adopted into our study without additional work).
> Also, note that our work provides templates (Figure 2) to ease the process of translating the inductive biases into prompts - an additional contribution of the FinePrompt framework (lines 76-77, 254-258).
>
> Transferring the fine-tuned inductive biases to prompts at this point demands manual annotation since the process requires a human to understand the central inductive bias in a model. Future work on fully automating the prompt construction step, we believe, is worth noting.
>
> >**`Question A` :** **“How are inductive biases obtained? Is it task-specific or instance-specific? Do inductive biases differ across different fine-tuned models for the same task? If not, this prompting strategy seems to be benefiting from some intermediate annotation of the task itself, rather than having anything to do with fine-tuning.”**
>
> The inductive biases studied in this work are obtained by transferring the main idea of fine-tuned models into prompts. Therefore, each prompt is model-specific and is neither task-specific nor instance-specific. Furthermore, the inductive biases differ among fine-tuned models since each model applies different kinds of model-specific inductive biases to solve a given task (e.g., DecompRC decomposes the multi-hop questions while SAE leverages graphs’ connectivity information to answer those same questions). Various kinds of models employ different fine-tuning strategies to obtain different inductive biases. Such inductive biases, the central contributing ideas of each model, are what’s being transferred to prompts.
>
> >**`Question B` :** **“Does the $P_{attr}$ you obtained for the numerical reasoning task vary depending on which few-shot examples are being presented to the model?”**
>
> Yes, the $P_{attr}$ (Equation 1; lines 105-107) obtained for numerical reasoning task depends on the few-shot examples and the input instance being presented to the model. This may result in sample variance, which was addressed by averaging our performance over 5 different iterations of randomly sampled $k$-shot samples (lines 173-182); averaging over multiple iterations is commonly applied to reduce sample variance in LLMs (Zhou et al., 2022).
>
> >**`Question C` : “In section 2.3 (Graph-Infused prompting) how are the nodes within the text identified? Are they manually identified or is this part of the dataset?”**
>
> We adopt the code base of the existing graph-based models (namely SAE, QDGAT) in identifying the nodes within the text. Therefore, it is neither manually identified nor a part of the MuSiQue dataset. While the methods of identifying the nodes in text may vary from model to model, we directly obtain these nodes based on the existing proposed graphs (SAE, QDGAT). Note that this point also emphasizes the transferability of our scheme on many different kinds of graphs used in previous works.
>
> >**`Question D` :** **“How are pipeline-infused prompts different from existing prompting strategies like CoT, self-ask, etc.? …”**
>
> While CoT, Self-Ask and other related approaches provide prompt-centric approches (e.g., dependent on well-curated, few-shot examples), the Pipeline-infused prompts are directly transferred pipelines of the previously validated fine-tuned models. For instance, DecompRC (Min et al., 2019) decomposes a given multi-hop question into three disparate types of questions (Bridging, Intersection, and Comparison Types) and chooses a set of single-hop questions identified by multiple decomposition types, based on which the model derives the final answer. In contrast, CoT entirely relies on human-curated few-shot samples of explanatory steps to get to the final answer, with each explanation demanding manual adjustment to whatever task it’s being used on (lines 110-114).
> >**`Question E` :** **“Is fineprompting done on GPT-4 out-of-the-box or one that's fine-tuned for each task?”**
>
> We do not fine-tune GPT-4 for FinePrompt. We simply use GPT-4 as a baseline to evaluate and validate our prompting strategy. Our work, given the abundance of existing fine-tuned inductive biases in the field, has a great potential to the research community because FinePrompt does not require the actual fine-tuning step while it benefits from the fine-tuned inductive biases.
>
> >**`Question F` :** **“As per your contribution list, can you explain how this method is more reproducible than others?”**
>
> The term “reproducibility” in line 74 is used to elaborate our point that the previously validated fine-tuned ideas can directly be “reproduced” in a form of prompts. Our objective is to reproduce the benefits of well-recognized and validated fine-tuned inductive biases from previous works in this new prompt learning era instead of inventing new, human-curated, prompting strategies.

---

### Official Review · Reviewer_fz2V · 2023-08-02

**Soundness:** 3

**Excitement:**

3: Ambivalent: It has merits (e.g., it reports state-of-the-art results, the idea is nice), but there are key weaknesses (e.g., it describes incremental work), and it can significantly benefit from another round of revision. However, I won't object to accepting it if my co-reviewers champion it.

**Paper Topic And Main Contributions:**

The paper proposes a method to construct prompts for compositional reasoning tasks based on transferring inductive biases from task-specific fine-tuned models to large language model (GPT4). The experimental results on MuSiQue dataset for MHQA and DROP dataset for numerical reasoning show that some types of the proposed prompts outperform existing elicitive prompting approaches.

**Reasons To Accept:**

The paper presents an interesting idea of improving large language models with insights from fine-tuned task-specific models and provides several experiments to support the claim.

**Reasons To Reject:**

- The claim in lines 148-149 about the performance of GPT-4 is not supported by any references.
- The proposed method is based on the inductive biases of the task-specific fine-tuned models, but the paper does not provide a comparison between different FinePrompt configurations and the original fine-tuned models based on which the prompts were created. Also, the general assessment of the FinePrompt method would benefit from the comparison with the current state-of-the-art methods on MuSiQue and DROP datasets.

**Reproducibility:**

5: Could easily reproduce the results.

**Reviewer Confidence:**

4: Quite sure. I tried to check the important points carefully. It's unlikely, though conceivable, that I missed something that should affect my ratings.

---

> ### Author Rebuttal · Authors · 2023-08-29
>
> Thank you for your constructive and thoughtful comments on our work. In the following, we would like to answer any questions you posed.
>
> > **“The claim in lines 148-149 about the performance of GPT-4 is not supported by any references”**
>
> The Technical Report by OpenAI (OpenAI, 2023) suggests that GPT-4 consistently exhibits superior performance in various fields, including textual compositional reasoning, compared to other LLM variants like GPT-3.5 and PALM. Although we have already cited this in our paper, we will also add the citation in Section 3.1 (lines 148-149) for clarity.
>
> >**“The proposed method is based on the inductive biases of the task-specific fine-tuned models, but the paper does not provide a comparison between different FinePrompt configurations and the original fine-tuned models based on which the prompts were created. Also, the general assessment of the FinePrompt method would benefit from the comparison with the current state-of-the-art methods on MuSiQue and DROP datasets.”**
>
> As for comparing different FinePrompt configurations, if you were referring to experiment settings such as zero-shot, few-shot ($k$=3) (lines 178-182), we first would like to emphasize that our work seeks to verify the improvement over the GPT-4 baselines (GPT-4, Self-Ask and CoT in Tables 1 and 2) with the injection of the fine-tuned inductive biases. By exploring both the zero-shot and few-shot settings in this work, we demonstrate that the introduction of the inductive biases effectively improves the compositional reasoning ability of GPT-4.
>
> Comparing against the fine-tuned models & SOTA models does not validate nor invalidate our approach, because: (i) we only need to compare against the base GPT-4 setting to verify if the inductive biases we extracted are useful or not, and (ii) the fine-tuned models are trained on the benchmark dataset (not a fair comparison to begin with), whereas the prompt-based approaches including FinePrompts utilize few-shot demonstrations only.
>
> We further elaborate on the details of choosing our models. The criterion for selecting the fine-tuned models in our study is whether they are well-recognized and validated by subsequent works in the field. We also focus on the reproducibility of the fine-tuned models, which is the reason the fine-tuned models used in our work are either SOTA on the official leaderboard (at the time of our submission) or are the ones available for experiments.

---

### Official Review · Reviewer_cDaD · 2023-08-05

**Soundness:** 3

**Excitement:**

3: Ambivalent: It has merits (e.g., it reports state-of-the-art results, the idea is nice), but there are key weaknesses (e.g., it describes incremental work), and it can significantly benefit from another round of revision. However, I won't object to accepting it if my co-reviewers champion it.

**Paper Topic And Main Contributions:**

The paper addresses the challenge of compositional reasoning in natural language processing, which involves understanding the meaning of complex sentences by combining the meanings of their constituent parts. The main contribution of the paper is the exploration of prompt templates as a way to transfer task-specific biases and improve the performance of GPT-4 on complex reasoning tasks. The paper presents experimental results that demonstrate the effectiveness of FinePrompt on multi-hop question answering and numerical reasoning tasks, and compares it to existing prompt techniques.

**Reasons To Accept:**

Firstly, it addresses an important challenge in natural language processing, namely compositional reasoning, which is a crucial aspect of language understanding.
Secondly, the paper proposes a novel approach to improving the performance of GPT-4 on complex reasoning tasks, using prompt templates to transfer task-specific biases. This approach has the potential to enhance the efficiency and effectiveness of natural language processing models.
Thirdly, the paper presents experimental results that demonstrate the effectiveness of FinePrompt on multi-hop question answering and numerical reasoning tasks, and compares it to existing prompt techniques. This provides valuable insights into the strengths and limitations of different prompt strategies.

**Reasons To Reject:**

This work does not exhaustively explore other finetuned models and LLMs, and that there is still room for improvement in incorporating other previously effective inductive biases into prompts to improve LLM reasoning ability.

**Reproducibility:**

4: Could mostly reproduce the results, but there may be some variation because of sample variance or minor variations in their interpretation of the protocol or method.

**Reviewer Confidence:**

3: Pretty sure, but there's a chance I missed something. Although I have a good feel for this area in general, I did not carefully check the paper's details, e.g., the math, experimental design, or novelty.

---

> ### Author Rebuttal · Authors · 2023-08-29
>
> Thank you for your valuable time and effort in reviewing our paper.
>
> > **“This work does not exhaustively explore other fine-tuned models and LLMs, and that there is still room for improvement in incorporating other previously effective inductive biases into prompts to improve LLM reasoning ability.”**
>
> Our study focuses on improving the compositional reasoning ability of GPT-4 in a textual understanding setting, specifically in numerical and multi-hop reasoning. Within the scope of these tasks, most models generally fall within the three categories specified in our work (Attribute, Pipeline, Graph). Furthermore, we aim to uncover the possibility of transferring the fine-tuned inductive biases of previously validated models to natural language prompts and verify if these biases help improve the compositional reasoning ability of GPT-4 in a textual understanding scenario (as demonstrated in Tables 1 and 2). We have elaborated on the concerns of incorporating different fine-tuned models and LLMs in our Limitations section (line 279-293), underscoring the infeasibility of exploring all available fine-tuned models and LLMs in the field. It is also worth noting that this is a short paper and exhaustively considering every single model that has been proposed to date is outside the scope of our work.

---

### Meta-Review · Area_Chair_YSBN · 2023-09-23

**Recommendation:** 3

**Metareview:**

the authors present an interesting approach to help bridge the gap between finetuning and prompt instructing of models, and show the effectiveness of their approach. The reviewers however agree on the lack of suffient rigour in the evaluation, especially wrt to alternative approaches to prompt tuning to better understand the gains and tradeoff with FinePrompt. The evaluation of GenBERT and SAE in the rebutal helps to address the main concern of a comparison with the finetuned models, and should be in the main text.
Still there is suffient material to make the paper interesting.  The comments of reviewer wpwG should be addressed in the final text to better clarify the work better.

---

### Decision · Program_Chairs · 2023-10-07

**Decision:**

Accept-Findings

**Comment:**

the authors present an interesting approach to help bridge the gap between finetuning and prompt instructing of models, and show the effectiveness of their approach. The reviewers however agree on the lack of suffient rigour in the evaluation, especially wrt to alternative approaches to prompt tuning to better understand the gains and tradeoff with FinePrompt. The evaluation of GenBERT and SAE in the rebutal helps to address the main concern of a comparison with the finetuned models, and should be in the main text.
Still there is suffient material to make the paper interesting.  The comments of reviewer wpwG should be addressed in the final text to better clarify the work better.